# How Physicians Renew Electronic Prescriptions in Primary Care: Therapeutic Decision or Technical Task?

**DOI:** 10.3390/ijerph182010937

**Published:** 2021-10-18

**Authors:** Taina Oravainen, Marja Airaksinen, Kaija Hannula, Kirsi Kvarnström

**Affiliations:** 1Clinical Pharmacy Group, Division of Pharmacology and Pharmacotherapy, Faculty of Pharmacy, University of Helsinki, 00014 Helsinki, Finland; marjaairaksinen@gmail.com (M.A.); kirsi.kvarnstrom@hus.fi (K.K.); 2Kirkkonummi Health Centre, 02400 Kirkkonummi, Finland; kaija.hannula@kirkkonummi.fi; 3HUS Pharmacy, University of Helsinki and Helsinki University Hospital, 00290 Helsinki, Finland

**Keywords:** electronic prescription, e-prescription, prescribing, optimizing prescribing, renewing, medication safety, medication management, population health, outpatient, primary care

## Abstract

In long-term pharmacotherapies, the renewal of prescriptions is part of the medication use process. Although the majority of medicines are used with renewed prescriptions, little research has focused on renewal practices. The aim of this study was to explore current renewal practices from a primary care physician’s perspective to identify system-based challenges and development needs related to the renewal practices. This qualitative study was conducted in two phases in public primary health care centres of Kirkkonummi, Finland. First, five physicians were shadowed on-site while they renewed prescriptions. The findings of the shadowing phase were further discussed in two focus group discussions with seven other physicians than in the shadowing phase. Inductive content analysis was used for data analysis utilizing Reason’s risk management theory as a theoretical framework. Due to problems in the renewal process, including impractical information systems, a lack of reconciled medication lists, and a lack of time allocated for renewing prescriptions, physicians felt that monitoring and reviewing each patients’ medications for renewal was complicated. Therefore, they felt that renewing, at times, became a technical task rather than a therapeutic decision. The physicians suggested information system improvements, enhanced interprofessional cooperation, and patient involvement as strategies to ensure rational pharmacotherapy and patient safety in the renewal of prescription medicines.

## 1. Introduction

Electronic prescribing (e-prescribing) has shown promise to enhance patient safety and medication management [1,2]. In electronic prescribing, a prescription is transmitted electronically from the physician’s office to the pharmacy. E-prescribing applications are already used in multiple countries worldwide [3,4,5,6]. In Finland, e-prescribing became mandatory at all levels of health care in 2017 [7]. In earlier studies, e-prescribing has been associated with improved workflow and efficiency [8,9,10,11,12,13,14] as well as easier monitoring of medication use and adherence [8,10,11,12,15]. In addition, e-prescribing has been shown to enhance patient safety, as electronic access to patient records and clinical decision support systems (CDSS) have reduced the risk of medication errors and have facilitated the management of medications [1,2,10,15].

Long-term pharmacotherapies should include regular evaluations and monitoring of the appropriateness of treatment [16]. This evaluation can be done at the point of renewing the prescription. Electronic renewing has been shown to ease monitoring medication adherence and possible overuse [8]. E-prescribing applications have also been found to speed up the processing of renewal requests [1]. Even though electronic renewing is a key functionality of most e-prescribing applications, little research has focused on challenges with electronic renewing, especially from a physician’s perspective [17]. According to the World Health Organization (WHO), more than 50% of medications are prescribed, dispensed, or sold inappropriately worldwide [18]. A systematic review found that a significant number of patient safety incidents in primary care were related to prescribing and medication management [19].

Renewing is rarely highlighted as an essential part of the medication use process [20,21] in ongoing efforts to improve the quality of care and rational use of medicines [18,22]. Still, primary care physicians can greatly affect the quality of care through their prescribing and renewal practices. Understanding how physicians perceive the renewing functionalities can help health care organizations develop more efficient and safer renewal practices as well as facilitate medication monitoring during renewal. In addition, physicians’ experiences can help allocate resources for necessary updates to e-prescribing systems. As chronic illnesses become more common [23], rational renewal practices become increasingly important.

The aim of this study was to explore the current renewal practices of electronic prescriptions from a primary care physician’s perspective and identify the factors affecting medication management, medication safety, and physicians’ workflows during renewals. In addition, this study investigated the physicians’ proposed solutions to improve renewal practices.

## 2. Materials and Methods

### 2.1. Study Context

In Finland, all prescriptions have been issued electronically at all levels of health care since the beginning of 2017 [7]. The precondition for prescribing or renewing a medication is that the physician has confirmed the need for the medication with a medical examination or using patient records [24]. In Finland, most healthcare units record patient data from their own information systems into the national Patient Data Repository [25]. The Patient Data Repository enables healthcare employees, not including community pharmacists, to access medical records recorded by other public and private healthcare units. Browsing the Patient Data Repository always requires a care relationship and the patient’s consent [25]. All electronic prescriptions (ePrescriptions) are stored in a nationwide centralized database called the Prescription Centre, which can be accessed by healthcare units and community pharmacies [7]. Viewing ePrescriptions requires the patient’s consent [7]. Physicians use a computerized order entry system integrated into their electronic patient record system (EPR) to write prescriptions. The EPR then sends the prescriptions to the National Prescription Centre. A paper or telephone prescription may be issued only during technical disruptions and in other exceptional cases [7]. ePrescriptions can be retrieved from the Prescription Centre and dispensed in any Finnish community pharmacy. When a prescription is dispensed in a pharmacy, the dispensing documentation is recorded to the ePrescription in the Prescription Centre [7]. The ePrescriptions and dispensing records are also electronically accessible to patients themselves through the My Kanta Pages [7]. In 2020, Finnish community pharmacies processed approximately 68.2 million ePrescriptions for a population of 5.5 million [26].

In Finland, prescriptions are mainly valid for two years [24]. The validity of prescriptions for central nervous system (CNS) and narcotic medications is restricted to a maximum of one year [24]. This means that prescriptions for long-term medications are usually renewed annually or every two years. In principle, all prescriptions can be renewed, with a few exceptions. Telephone prescriptions cannot be renewed, and physicians may prevent the possibility of renewal of a prescription at their discretion [24]. Patients can request a renewal of an ePrescription while visiting a doctor, or by contacting a healthcare unit or their personal physician, if they have one, by phone [7,27]. Patients can also send a renewal request electronically online via the My Kanta Pages, or a community pharmacist can send the electronic request with the patient’s consent [7]. Electronic renewal requests are sent to the Prescription Centre, from which it is forwarded to the designated healthcare unit [27]. This does not have to be the same healthcare unit where the prescription was originally issued, and not all Finnish healthcare units accept electronic renewal requests [27]. After receiving the renewal request, the healthcare unit passes the request on to a physician. The patient or the community pharmacist sending the electronic request cannot influence which physician processes the renewal request [27]. Electronic renewal requests must be processed within eight days after receiving them [24]. A physician can either accept, deny, or return a renewal request when processing it [27]. If the request is not processed within eight days, the Prescription Centre will mark it as lapsed.

### 2.2. Data Collection

In this study, we wanted to focus on ePrescriptions and renewals issued by primary health care physicians to outpatients. The study was carried out in the Kirkkonummi Health Centre, which provides municipal health services for the 40,000 inhabitants of Kirkkonummi, a municipality located in the metropolitan area of Finland. In Kirkkonummi, every resident has a designated personal physician that provides their primary health care. Not every municipal health centre in Finland provides personal physicians or family medicine doctors for their residents. The Kirkkonummi Health Centre consists of three health centres—the main health centre in Kirkkonummi City Centre and two smaller health centres in Veikkola and Masala. An electronic prescribing software called Pegasos was used in the Kirkkonummi Health Centre during the time of the study. The Pegasos software contained an integrated drug–drug interaction alert system that created alerts about possible interactions between medicines on the EPR’s medication list and the medicine being renewed. The alert system created the interaction alerts based on the drug interaction database called Inxbase. Drug allergy alerts or adverse drug reaction alerts were not included in the alert system used in Kirkkonummi. The data collection was carried out between April 2019 and July 2019. All participating physicians were recruited by email invitation with purposive sampling, and the medical director of the Kirkkonummi Health Centre helped with the recruitment process.

A qualitative design was chosen for this study since few studies have investigated the renewing of prescriptions, particularly electronic renewing, globally. We wanted to gain a comprehensive picture of renewing as a process and to explore factors that affect physicians’ workflows and the safety of patients during renewing. Therefore, in this study, we combined two qualitative study methods—on-site physician shadowing and focus groups (Figure 1). Only primary care physicians were recruited for this study, as we sought to understand the physicians’ insight into current renewal practices and how the practices affect the physicians’ work.

In the first phase, we utilized shadowing (Figure 1), which is a research method in which a research subject is observed over a period of time [28]. Shadowing can provide detailed data about the mundane aspects of the work processes that can be difficult to articulate with other study methods [28], for example, surveys and interviews that have been carried out in earlier studies [8,9,10,11,12,29]. By utilizing the shadowing method, we wanted to obtain in-depth information about the everyday work practices related to the renewal of prescriptions at one Finnish health centre.

First, the participants for the shadowing phase were recruited and informed about the study. The participating physicians were shadowed on-site by a researcher (TO) while they renewed prescriptions. Findings and observations about the renewal practices were gathered and documented as field notes. The field notes were handwritten on a pre-made observation form to facilitate structuring the observations. The observation form was used to gather and document observations about the various stages of the renewal process, the monitoring of medications, the reviewing of the patient’s medication regimen, the use of information systems, and the communication between the GP and the patient as well as between different healthcare providers while making the renewal decision. In addition, the time spent shadowing and the number of renewal requests handled by the physician during the shadowing period was recorded on the form. The observation form was developed based on the study objectives, and also considered the theoretical framework and the regulations for the renewal of medications.

To obtain more comprehensive data, the physicians were encouraged to “think aloud” about the issues they took into consideration while making their renewal decisions. If necessary, TO asked questions for clarification or to understand the reasoning behind the physicians’ decisions. The answers to these questions were also documented in the field notes. Each physician was shadowed once for a time range of 31–73 min. Before analysis, the handwritten field notes were recorded to an MS Word document.

After completing the shadowing phase, the physicians for the focus group discussions (FGDs) were recruited (Figure 1). Two focus group sessions were conducted, both at the main Kirkkonummi Health Centre. The physicians who participated in the FGDs were different from those shadowed in Phase 1 of the study. The semi-structured questions for the focus groups were designed based on previous literature [10,11], and the shadowing data were analysed before the interview guide was established (Figure 1). During the FGDs, the interview guide was followed, despite topics that were spontaneously and comprehensively discussed within the group. Both focus group discussions lasted for 60 min, and they were audiotaped and transcribed verbatim to an MS Word document.

### 2.3. Data Analysis

The shadowing and focus group data were analysed by inductive content analysis. The theoretical framework of this study was James Reason’s theory on human errors, viewed from the systems approach [30]. During the analysis, meaningful expressions relating to the research objective were searched from the data and coded to emerging categories and themes. The emerged themes and categories were then grouped into main categories. The analysis units were single words, sentences, or groups of sentences. Qualitative analysis software Atlas.ti (version 8.4.15.0) (ATLAS.ti Scientific Software Development GmbH, Berlin, Germany) was utilized in the analysis. TO carried out the primary coding and analysis of the data. All authors read the transcripts and results, and the conclusions were confirmed through discussion.

### 2.4. Ethical Statement

The study was carried out in accordance with good scientific practice and national research ethics guidelines [31]. The study received a study permit from the review board of welfare services of Kirkkonummi. Ethical approval from the hospital district’s ethics committee was not applied since neither patients nor prescription records were reviewed during the study, and the physicians’ decisions were not influenced by the researcher (TO). All participants gave their written informed consent to take part in the study and to be audiotaped during the focus group discussions. The participants were reassured that participation in the study was voluntary and could be cancelled at any time.

## 3. Results

A total of 12 primary health care physicians participated in the study. Of these, five physicians participated in the shadowing phase and seven physicians participated in the focus group discussion phase. Of the participants, 42% (*n* = 5) were female, nine participants (75%) were general practitioners, and three (25%) were specialists in general medicine. On average, participants were 37 years old (age range of 27–58). There were participants from all three health centres of Kirkkonummi.

### 3.1. The Renewal Process

In the Kirkkonummi Health Centre, the renewal of ePrescriptions is a multi-stage process. The stages of the renewal process are presented in Figure 2. Generally, the physicians were content with the personal physician model and the technical implementation of the renewal process. According to the physicians, renewing was seen as both a therapeutic decision and a technical task. The physicians wanted to take the therapeutic aspects into account when renewing medications, but renewing became a more technical task when they had multiple renewal requests to process within a limited time.

The physicians brought up patient-related and medication-related factors that facilitated renewing. Renewing was easier to complete when the physician knew the patient, as they had a better understanding of the patient’s medication and care matters and could renew prescriptions for longer periods than for unknown patients. Renewing was also easier when patients had regular follow-up visits with a doctor or a nurse, had regular laboratory controls, or had health appointments in secondary care as well, for example, with a geriatrician or a specialist at the contraception clinic who monitored their health and medication. Even high-risk medications, such as strong pain medications and warfarin, could be renewed for these patients without asking the patient for a specific check-up. Some participants felt that if they did not know the patient, they could not renew certain medications, such as CNS medications, narcotics, and other medications with potential for abuse. Renewing was also difficult when a patient did not attend their regular check-ups.

The participants felt that some medications were “easier” to renew than others. This was because physicians considered some medications as “harmless”, “safe”, or “kind”, and thus, there was little need for monitoring while renewing. For example, the physicians mentioned melatonin, antihistamines, and paracetamol (acetaminophen) among these medications. In addition, prescriptions for long-term medications were considered “easier” to renew than those for short-term medications or those to be taken if necessary.

### 3.2. Challenges with the Renewal Process

#### 3.2.1. Therapeutic and Communicational Challenges

The physicians raised multiple problems that they had encountered in the renewal process (Figure 2, Table 1). There were usually a lot of renewal requests to process and there was not enough time allocated to check and monitor medications comprehensively while renewing prescriptions. The physicians felt they often received renewal requests in an uncoordinated manner, which slowed down renewing; renewal requests from unknown patients and those concerning prescriptions that had been issued at different health centres or in secondary care were perceived as especially problematic. Usually, the physicians had to study the patient’s medical history from the EPR and check their previous laboratory controls, current prescriptions, and previous dispensing events from the Prescription Centre, but some requests had to be declined if the GP did not know how the medication should be monitored and there was not an explicit treatment plan. The physicians felt that making renewal decisions about unknown patients’ medications or about medications the GPs had little clinical experience of prescribing was complicated, and they had to ponder if it was riskier to renew the medication without necessary monitoring or to leave the patient without the medication.

The participants felt some renewal requests were unnecessary because the medications were meant to be used for the short-term or had been discontinued and they could not find an up-to-date treatment plan or a valid reason to continue the medication, for example, antibiotics, muscle relaxants, and benzodiazepines. Processing these renewal requests usually required the physicians to search for additional information from the patient records or to contact the patient. Within the current renewal system, patients were not able to provide the physician with any additional information in the renewal request.

Recent medication shortages in Finland also hindered and slowed down renewing according to the physicians. Information about medication shortages was not maintained in the EPR, and finding an alternative medication increased the physicians’ workloads. The physicians usually received information about medication shortages from community pharmacists or their peer physicians.

#### 3.2.2. Technical Challenges

Monitoring medications was complicated by technical difficulties with patient records. Finding necessary information and getting a comprehensive picture of the patient’s health status and care was laborious since the records were divided into different electronic databases, for example, the EPR and the Patient Data Repository, and the information was presented incoherently. The medication list in the EPR was rarely up to date because medications prescribed at other healthcare units are not added to the list automatically. It also often contained unnecessary medications, for example, duplicate medications. Because of this, the medication list in the EPR was not always seen as a reliable and explicit information source during renewing. In addition, the participants felt it was problematic that after stopping medications or making other changes in the medications in the EPR, the changes did not update automatically in the Prescription Centre and vice versa.

The physicians also felt that the Prescription Centre was not a reliable source for reviewing the patient’s medication. According to the physicians, the Prescription Centre often contained unnecessary and duplicate prescriptions. A significant problem with the Prescription Centre was also its technical clumsiness and disorganized data. The physicians complained that the dispensing documentation did not appear in an understandable format, which hindered checking on adherence and possible overuse. Using the Prescription Centre was usually slow, especially if the patient had several ePrescriptions.

All in all, finding out and reviewing the patient’s full medication regimen was seen as difficult and time-consuming with the current patient information systems. The physicians felt that getting a completely realistic and clear understanding of the patient’s entire medication regimen was rarely possible. When determining the medications actually in use, the physicians sometimes had to ask the patient over the phone or during a control visit. However, some patients were unable to name their medications upon request. These system-based challenges posed multiple patient safety risks in the renewal process, as illustrated in Figure 3.

Conversely, the physicians were mostly satisfied with the interaction alerts they received when renewing prescriptions because they helped detect harmful drug–drug interactions. However, they still criticized the volume and low clinical significance of most of the alerts. Many physicians felt that other than the most critical interaction alerts, most were unnecessary. They had also received interaction alerts about medications that had been marked as paused or had already been stopped and removed from the patient’s medication list.

There was little cooperation between the Kirkkonummi Health Centre and local community pharmacies, and, usually, the physicians communicated with the local community pharmacists when pharmacists called and asked for clarification or modifications for prescriptions. The physicians were at times frustrated with the community pharmacies’ practices, and they had faced problematic situations concerning the pharmacists’ medication counselling and medication shortages.

### 3.3. Development Proposals to Improve the Renewal Process

The participants brought up multiple strategies to improve the renewal process (Table 1). The development proposals involved better coordination of renewing, up-to-date medication lists and treatment plans, patient involvement, and interprofessional cooperation.

The physicians felt renewing should rather be done during physician appointments, as it would facilitate monitoring medications. This was seen as particularly helpful with older patients and patients with multiple illnesses. Synchronizing all renewals of long-term therapies to be done at the same time would also reduce the physicians’ workload.

According to the physicians, the medication list in the EPR should be connected to the Prescription Centre so that medications prescribed and electronically sent to the Prescription Centre would automatically be added to the EPR’s list as well. This was seen as an improvement to safety during renewing. One of the participants thought that renewing would also be easier if up-to-date treatment plans for high-risk medications were maintained in the EPR.

The physicians wished for interaction alerts to be customizable and to contain more information. Firstly, the participants wished that the medication alerts were less repetitive and that certain alerts could be marked as processed per doctor or per patient, either permanently or for a fixed period. Secondly, the participants suggested the alerts should contain information about alternative medications to avoid drug–drug interactions. They also wished to receive alerts based on the medications in the Prescription Centre in addition to the medications in the EPR.

According to the participants, the layout of the Prescription Centre should be more visual so that dispensing events could be reviewed more quickly. One physician also suggested that the Prescription Centre would automatically count how much the patient has used their medication, based on the dosage regimen and the dispensing records. These improvements would ease monitoring medication use. In addition, the physicians wished the Prescription Centre could be used as a medication list, and it would be mandatory for every healthcare unit to keep it up to date. Cancelling unnecessary prescriptions was seen as vitally important in maintaining medication lists. Actively cancelling unnecessary prescriptions, for example, during appointments, would also clarify the prescription list the patient sees in the My Kanta Pages and could help reduce unnecessary renewal requests. In addition, the role of the patient in maintaining the medication list was highlighted. The participants suggested that patients could mark the medications they are currently using through the My Kanta Pages. Patients could receive a reminder via SMS to review and verify their medication list through the My Kanta Pages.

The physicians also suggested new interprofessional operating models for reviewing medication lists in the Kirkkonummi Health Centre. In one model, nurses would review and update the medication list with the patient once a year. These medication list reviews would free up the physicians’ time for other work tasks and relieve the mental workload. According to the physicians, medication list reviews could also be done by community pharmacists during the patients’ pharmacy visits. Overall, the physicians wished there was more cooperation with the local community pharmacists, and they suggested holding joint meetings with community pharmacists.

## 4. Discussion

### 4.1. Main Findings

The physicians were well aware that renewing a prescription is a therapeutic decision, but multiple system-driven problems and limited time allocations made it a rather technical task. The major problems in the renewal process were related to the receiving of uncoordinated renewal requests, having incoherent, disorganized, and outdated information in the EPR and in the Prescription Centre, which led to outdated medication lists and made it difficult to review the entire medication regimen for a renewal decision.

Our study revealed several impracticalities in the electronic patient information systems and the computerized order entry system that made renewing difficult and inefficient. Physicians have stated in previous studies that e-prescriptions and electronic renewing have made processing renewals faster and simpler [1,29]. However, the current information systems are not optimized for therapeutic decision making when renewing prescriptions. According to our study participants, they had the premises for medication reconciliation and monitoring, but in practice, this work was often technically too laborious because the patient’s current medication regimen was not readily available. The physicians estimated that with the help of the Prescription Centre they could only get “an educated guess” of the patient’s medication. The e-prescribing application and the EPR had multiple impractical features, problems, and shortcomings due to which reconciling and reviewing the patient’s overall medication and monitoring medication use was hindered. The physicians were frustrated, especially because of the poor interoperability between the EPR and the Prescription Centre. The medication lists in the EPR and in the Prescription Centre were not congruent, as medication information and medication changes are not transferred between EPRs at different levels of public and private health care [32,33] nor between the EPRs and the Prescription Centre. The physicians also did not always have time to reconcile the medication lists, which contributed to the fact that the lists were rarely up to date. Similar issues in the reconciling of patients’ medications have been reported in previous studies in Finland [10,11,33] and in the United States [34]. Because of the poorly up-to-date medication lists, it is difficult for any physician to take comprehensive responsibility for a patient’s pharmacotherapy [32].

The lack of up-to-date medication lists and challenges in monitoring medications posed patient and medication safety risks in the renewal process. In addition, a lack of time allocated for processing renewals, and renewal requests arriving in an uncoordinated manner complicated renewing. Despite the challenges, the patient could often not be left without medication. Physicians had to weigh the safety risks associated with accepting and denying renewal requests. The Finnish Ministry of Social Affairs and Health (MSAH) has stated that renewing should be a well-planned part of a patient’s care, especially with older adults, and the need, safety, and implementation of medication should be reviewed annually [20]. Healthcare professionals should also utilize the patient’s own assessment to monitor medications [35], but currently, medication monitoring data recorded by the patients themselves are poorly available. In practice, implementing the recommended procedures was too laborious. Because of the incomplete information, the participants felt that a comprehensive medication review and medication changes should not be done while renewing prescriptions. Instead, medication monitoring was preferably done during appointments, as was also stated in a previous Finnish study [29]. Still, the physicians felt that there is not enough time to deal with medication problems or to go through all the necessary matters even during appointments [32,33]. Renewing should be an integral part of the pharmacotherapeutic process, and health care organizations should be given concrete recommendations on how renewing should be organized as well as how to improve safety in the renewal process.

The framework of the renewal process is largely based on the Finnish Electronic Prescription Act [7]. Thus, in our study, the renewal process proceeded in several stages in a similar way as in a previous study [29]. The purpose of the Prescription Centre was to make it possible to establish the patient’s overall medication regimen so that it could be considered when planning and implementing pharmacotherapy [7]. Based on the physicians’ experiences, these objectives were not met in practice. The Prescription Centre is used nationwide in Finland, and Finnish physicians have criticized the disorganized layout and technical problems, such as slowness and system crashes, of the Prescription Centre earlier [10,11,29]. In one study, physicians stated that they felt that they receive no help from the Prescription Centre during the renewal process [10]. The participants of this study also criticized several technical issues of the EPR and the Prescription Centre. Clumsy features and technical problems are not rare in e-prescribing systems. In the United States, physicians have also reported experiencing problems with network connectivity, system glitches, clumsy features, and pharmacies not reliably receiving e-prescriptions [8,13,14,34]. Technical issues usually require extra time to resolve [13], and this usually reduces the time available for medical care. In this study, the participants wished that the dispensing records and the patient’s medication use would be presented unambiguously in the Prescription Centre, as it would help determine medication adherence. E-prescribing applications and ERPs should facilitate making informed therapeutic decisions and support the physician. In addition, the usability of the systems, from the physicians’ point of view, should be better considered when developing the systems.

Steps have already been taken to improve patient safety and ease physicians’ workloads. The physicians felt that the clinical decision support system with drug interaction alerts in the EPR was useful and facilitated noticing severe drug interactions. Similar experiences have been reported in previous studies [34,36,37,38]. In addition, a nationwide medication list will be included in the Prescription Centre in the coming years [39]. The medication list will include all medications a person is taking, and it is updated automatically whenever medication data is recorded in the Prescription Centre, so it does not require separate updating. Citizens will also be able to access the up-to-date medication list through the My Kanta Pages [39].

The participants wished there was more interprofessional cooperation and patient involvement in the renewal process. They suggested that nurses and pharmacists could review patients’ medications at the health centre or community pharmacies. GPs have earlier suggested that nurses or pharmacists would update patients’ medication lists before GP appointments [33]. The participants stated that the current cooperation with local community pharmacists was limited. Previous studies have reported that there is not extensive enough cooperation between health centres and community pharmacies in the renewal process [29] and in managing medication-related risks [40]. Therefore, other professions’ expertise is not utilized sufficiently [32]. The physicians in Kirkkonummi were open to increasing cooperation and suggested holding joint meetings with local community pharmacists. Familiarizing themselves with each other’s practices and IT systems would help to develop cooperation between physicians and pharmacists [29]. In addition, there is a need to increase cooperation between physicians at different levels of health care. The participants hoped that treatment plans and medication lists would be updated collaboratively, as it would simplify renewing and monitoring medications prescribed in secondary care. The physicians also highlighted the importance of actively cancelling unnecessary and duplicate prescriptions. Finnish community pharmacists may also cancel ePrescriptions, but it is considered to be the responsibility of physicians to cancel prescriptions because community pharmacists do not have access to medical records. MSAH has also emphasized drawing up and maintaining treatment plans [21,35,41]. Interprofessional approaches and shared responsibility for the accuracy of information about patients’ medication regimens would ease the physicians’ workload in the renewal process.

In addition, the participants hoped patients would take a more active role in self-managing their medications, as it was sometimes difficult to evaluate the effectiveness of the pharmacotherapy and the patient’s adherence using only databases. Patients’ adherence to treatment is increasingly important to follow, as the validity of prescriptions has been extended to two years and contact with physicians can be less frequent than before. At the same time, patients not knowing their medications has become a recurring problem, as indicated in this and in previous studies [33]. Medication users should have a good understanding of the correct way to use their medications as well as their treatment plan and goals [21,41]. This can be achieved with a partnership between physicians and patients and supporting patients to take more responsibility for their pharmacotherapy [21]. In a previous study, physicians stated they were open to working more as a coach rather than as an authoritative figure [33].

### 4.2. Limitations and Strengths

All 12 participating physicians of this study were recruited from the same medium-large health centre in Finland, which can limit the generalizability of the findings. Due to the personal physician model used in the Kirkkonummi Health Centre, physicians had known many of their patients for a long time. As a result, medications may be better managed in Kirkkonummi than in health centres not applying a personal physician model. In addition, the participants might have deliberated their decisions more carefully during shadowing because of the Hawthorne effect [28]. Similar experiences of the renewal process were obtained and repeated in the shadowing phase and in the focus group discussions, but because of the small sample size, saturation of the data was not obtained. By implementing two qualitative study methods, we expected to obtain sufficient observations and data on the renewal practices with a sample size of 12 physicians. The sample size was considered to be sufficient to represent the views of the physicians of the Kirkkonummi Health Centre, as the participants were both general practitioners and specialists in general medicine, and they were of different ages and from all three health centres of Kirkkonummi. With more extensive data, more views may have been found on the research questions and the reliability of this study could have been improved. However, as electronic renewing is not a well-researched topic, we believe the findings of this study provide useful insight into the challenges and development needs of the renewal process.

Despite the small sample size of this study, the findings were similar to those from previous studies. Similar renewal practices and factors affecting renewing were reported in a previous Finnish study [29]. Additionally, difficulties identified regarding medication management and the lack of up-to-date medication regimen information were in line with previous publications [10,11,21,29,32,33,35,41]. Experiences with the poor usability of the e-prescribing applications from the physicians’ point of view have been reported in studies conducted in other countries. By combining two research methods, we aimed to obtain diverse data and proposals for solutions, not only pitfalls. Analysing the shadowing data before the focus group discussions helped us focus on the topics and problems raised by the physicians themselves.

The purpose of this mixed-method qualitative study was to present the perspective of primary care physicians on the challenges and development needs of the renewal process. Understanding the physicians’ insight on renewing can make it easier for researchers and health services planners to design new interprofessional approaches and interventions to develop the renewal process. In addition, the findings of this study may help policymakers allocate resources for the necessary updates to the Prescription Centre.

## 5. Conclusions

Impractical information systems and technical problems, poorly up-to-date medication regimen information, problems in information flow, and a lack of time pose challenges in the renewal process. According to the participants, reconciling the patient’s entire medication regimen and medication monitoring is laborious during renewing. The impractical information systems hindered making informed therapeutic decisions, and with limited time allocated for renewing, it became more of a technical task. According to the participants, the renewal process could be improved with better coordination of renewing, information system improvements, better interprofessional communication and cooperation, and involving the patient in taking more responsibility for their medications. Renewing should be seen as an integral part of the medication use process, and health care organizations should be given comprehensive guidance on how patient safety and the rational use of medicines could be ensured in the renewal process.

## Figures and Tables

**Figure 1 ijerph-18-10937-f001:**
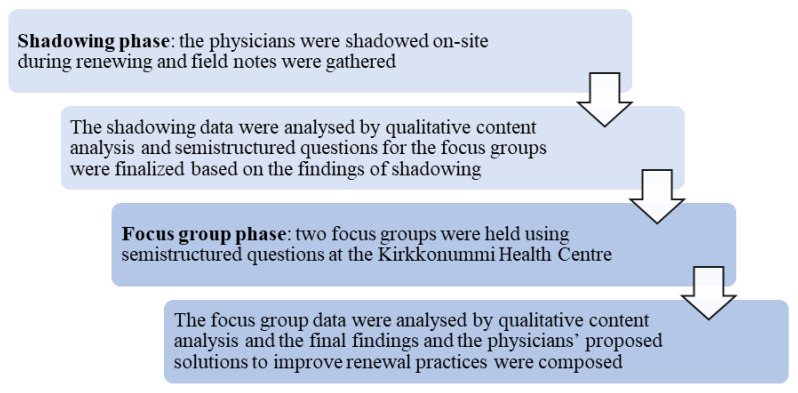
The two-phase study design.

**Figure 2 ijerph-18-10937-f002:**
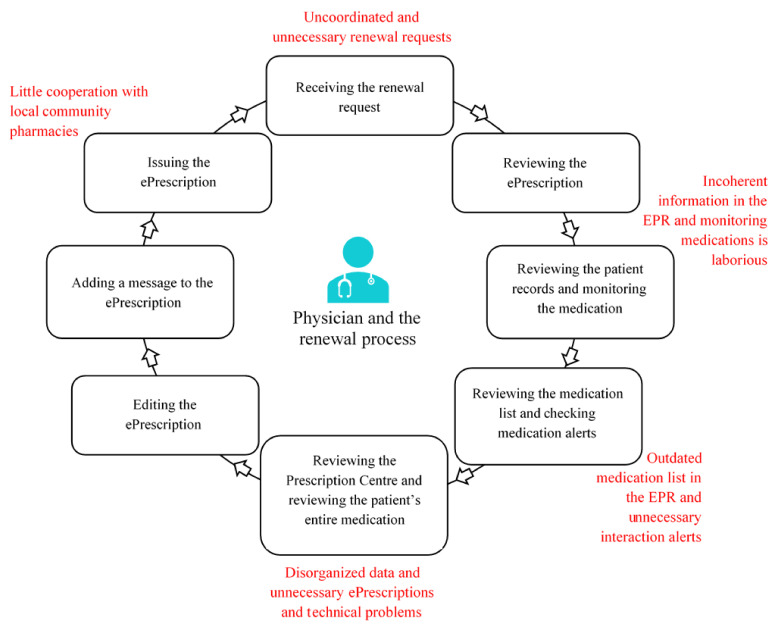
The stages and challenges of the electronic renewal process, constructed based on the two-phase data collection. EPR = electronic patient record system.

**Figure 3 ijerph-18-10937-f003:**
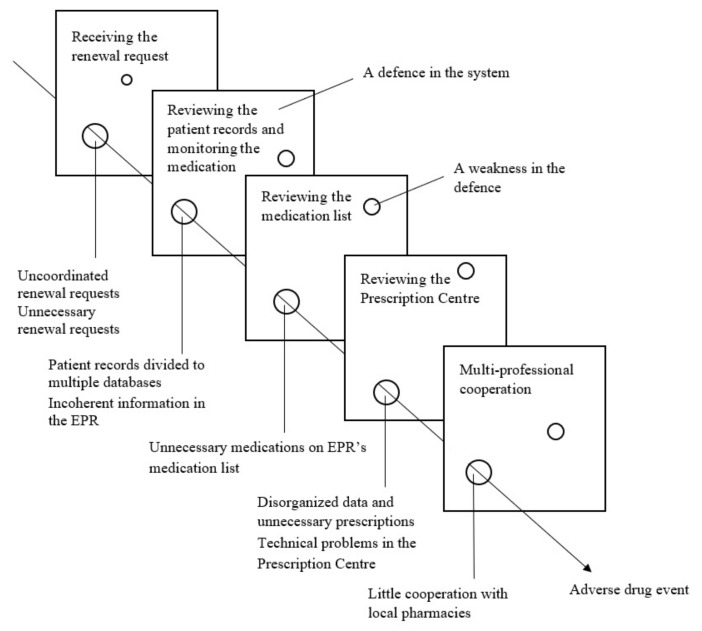
Patient safety risks identified in the renewal process and illustrated with Reason’s Swiss Cheese Model [30]. EPR = electronic patient record.

**Table 1 ijerph-18-10937-t001:** The physicians’ complaints and development proposals concerning the renewal process. EPR = electronic patient record system.

Complaints	Development Proposals
Receiving the renewal request	
Uncoordinated renewal requests Unnecessary renewal requests	Renewing during appointments
Reviewing the patient records and monitoring the medication	
Monitoring medications is laborious Patient records are divided into multiple databases Information in the EPR is presented incoherently	Maintaining an up-to-date treatment plan
Reviewing the medication list and checking medication alerts	
EPR’s medication list is rarely up to date Unnecessary medications on EPR’s medication list Volume and low clinical significance of interaction alerts Unnecessary interaction alerts	Updating the medication list automatically based on the Prescription Centre Extending interaction alerts to prescriptions in the Prescription Centre Marking alerts as processed Providing information about alternative medications to avoid interactions
Reviewing the Prescription Centre and reviewing the patient’s entire medication regimen	
The Prescription Centre is not a reliable source for reviewing the patient’s medication Reconciling and reviewing the patient’s entire medication regimen is difficult and laborious Disorganized data and unnecessary prescriptions in the Prescription Centre Having to cancel prescriptions from the Prescription Centre separately Technical problems with the Prescription Centre Patients’ poor knowledge of their medications	Maintaining up-to-date medication lists in the Prescription Centre Actively cancelling unnecessary prescriptions Explicit documentation of dispensing events in the Prescription Centre Medication list reviews conducted by nurses Medication list reviews conducted by pharmacists Medication list verification request for patients
Interprofessional cooperation	
Frustration with community pharmacy practices Little cooperation with local community pharmacists	Joint meetings with community pharmacists

## Data Availability

The data obtained and analysed during this study are not publicly available but are available from the corresponding author upon reasonable request.

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
