# Peer review of "How Physicians Renew Electronic Prescriptions in Primary Care: Therapeutic Decision or Technical Task?"

_ijerph, 2021, doi:10.3390/ijerph182010937_

Round 1

Reviewer 1 Report

Very interesting paper to read, however I have a few points I would like authors to clarify:

  1. Please rephrase the following sentence

 Of studies included in a literary review, 48% indicated that e-prescribing applications also speed up processing renewal requests [1].

  1. Please eyplain in greater detail the following In Finland prescriptions are mainly valid for two years [24]. Validity of prescriptions for central nervous system (CNS) and narcotic medications is restricted to a maximum of one year – does this one and two years refer to the renewal time i.e. 12 times/24 times or is it possible for a single prescription to be valid for 2 years, this doesn't make any sense

  1. The patient or the community pharmacist sending the request cannot influence which physician processes the renewal request. – do patients have family medicince doctors?? Further you state that they do, does the family doctor renew the prescription or is there another system by which any doctor can access the patient data an renew a prescription

  1. How was this developed pre-made observation form

  1. Who were members of the focus groups? Only physicians? Please describe in greater detail

  1. Please provide a rational for the low sample size

  1. Electronical monitoring of prescriptions allows for control over adherence of patients i.e. if a patients collects only one but not the other drug... can something be said about this?

  1. For readers... please explain whether there is a pharmacy based monitoring of patients treatments, is there an option in the system as an aleret if i.e. benzodiazepines are prescribed longer than 3 months etc...also please explain who regulates if the drugs can be prescribed for renewal, the drugs agency? Healthcare provider? Is this regulated? It is confusing that antibiotics and benzodiasepines can be renewed as this is not the usual case

  1. Monitoring medications was complicated by technical difficulties with patient records. Finding necessary information and getting a comprehensive picture of the patient’s health status and care was laborious since the records were divided to different electronic databases and the information was presented incoherently. – who can access these records? Only GPs? Emergency doctors? Specialist? Pharmacists? Do patients have any control over who they allow to view their records? Please explain

  1. Double prescriptions-is this something a pharmacists could correct? Do pharmacists have access to all the records?

  1. The results of this study are not easily generisable as it referes strictly to the system in Finland

  1. Do try to include more references of a newer date.

  1. What happens when there is no internet connection? There are absolutely no paper prescriptions?

  1. How are the „techincal“ jobs handled in other countries? Can GPs have additional staff to handle the „boring“ work?

Reviewer 2 Report

Dear authors, it has been a pleasure to read your manuscript about the electronic prescription system in Finland and how you have investigated care physicians perspectives on the system.

I only have very few minor comments you can consider.

Minor comments

  • line 35. Is electronic prescription mandatory for all prescriptions from general practice, hospital, dentist, veterinarian...? What are the rules? Shortly described in methods. Please consider if the introduction is a better place.
  • Renewal. The process is a bit unclear. Can a patient not ask their personal GP to renew a prescription? 
  • line 139. Referring to previous literature please add references
